

# Symmetry restoration and the gluon mass in the Landau gauge

Urko Reinosa[1], Julien Serreau[2], Rodrigo Carmo Terin[3,4] and Matthieu Tissier[4]

**1** Centre de Physique Théorique (CPHT), CNRS, École Polytechnique,
Institut Polytechnique de Paris, Route de Saclay, F-91128 Palaiseau, France.
**2** Université de Paris, CNRS, Astroparticule et Cosmologie, F-75013 Paris, France.
**3** Universidade do Estado do Rio de Janeiro (UERJ), Instituto de Física,
Departamento de Física Teórica, Rua São Francisco Xavier 524,
Maracanã, Rio de Janeiro, Brasil, CEP 20550-013
**4** Sorbonne Université, CNRS, Laboratoire de Physique Théorique de la Matière Condensée,
LPTMC, F-75005 Paris, France

## Abstract

We investigate the generation of a gluon screening mass in Yang-Mills theory in the Landau gauge. We propose a gauge-fixing procedure where the Gribov ambiguity is overcome by summing over all Gribov copies with some weight function. This can be formulated in terms of a local field theory involving constrained, nonlinear sigma model fields. We show that a phenomenon of radiative symmetry restoration occurs in this theory, similar to what happens in the standard nonlinear sigma model in two dimensions. This results in a nonzero gluon screening mass, as seen in lattice simulations.


doi:10.21468/SciPostPhys.10.2.035

# 1 Introduction

The similarities between the nonlinear sigma (NL$\sigma$) model in $d = 2$ spacetime dimensions and Quantum Chromodynamics (QCD) in $d = 4$ have been reported long ago. Both theories exhibit asymptotic freedom [1–4]. Moreover, their low-energy excitations are gapped, while their microscopic descriptions involve massless fields. In the case of the NL$\sigma$ model, this apparent change of spectrum is a consequence of the radiative restoration of a symmetry as we now recall in the simplest O($N$)/O($N-1$) case. The NL$\sigma$ model can be interpreted as describing the ordered phase of a $N$ component vectorial model, where the radial fluctuations are frozen. The theory therefore involves $N - 1$ massless Goldstone modes. However, the Coleman-Mermin-Wagner theorem [6,7] states that no such ordered phase exists in $d = 2$. The true spectrum of the theory involves, instead, $N$ degenerate massive modes, with an exponentially small mass $m \sim \mu \exp(-\text{const.}/g)$, with $\mu$ an ultraviolet (UV) scale and $g$ the (running) coupling constant [5]. This symmetry restoration phenomenon is best understood in the Wilson functional renormalization-group (RG) approach, where one follows the evolution of the effective potential as modes above a RG scale $k$ are progressively integrated out [8]. For $k$ of the order of the microscopic scale of the theory, one starts with an O($N$)-symmetric potential, strongly peaked around a nonzero value, which ensures that the radial modes are frozen and that only the transverse, pseudo-Goldstone degrees of freedom (d.o.f.) contribute. These massless modes lead to large infrared fluctuations which, in $d = 2$, are strong enough to result in an effective symmetry restoration: the minimum of the effective potential decreases with decreasing $k$ and eventually vanishes below some scale $k_r$. One can picture that regime as an incoherent collection of regions of broken symmetry of size $1/k_r$. On infrared scales, the symmetry in internal space is effectively restored and all modes are massive and degenerate.

A somewhat similar situation occurs in Yang-Mills theories, where, despite the fact that the microscopic d.o.f. of the theory (the gluons) appear massless on UV scales, the long distance, physical excitations are massive glueballs. Not only that: it is, by now, well-established [9–12] that, in the Landau gauge, the gluon propagator actually reaches a finite nonzero value at vanishing momentum, corresponding to a nonzero screening mass. It is not clear at the moment whether this results purely from the nontrivial infrared dynamics between the massless degrees of freedom of the Faddeev-Popov (FP) action, or if it originates from a deformation of the latter due to the gauge-fixing procedure on the lattice, in particular, the way the Gribov ambiguity is dealt with [13–15].

In Ref. [16], a gauge-fixing procedure was proposed in the Landau gauge, where Gribov copies are averaged over with a nontrivial weight, which provided an intriguing novel connection with the physics of the NL$\sigma$ model described above. First, the continuum formulation of this procedure involves a set of auxiliary NL$\sigma$ fields and, second, the existence of supersymmetries in the NL$\sigma$ sector effectively reduces the number of spacetime dimensions by 2, a phenomenon similar to what happens (for sufficiently large number of dimensions) in disordered systems in statistical physics, known as the Parisi-Sourlas dimensional reduction [17]. Just like the Higgs' phenomenon [18], the coupling of these constrained NL$\sigma$ fields to the gluon gives a mass to the latter. The main point of Ref. [16] was to actually explore the possibility to explain the observed gluon mass on the lattice in this way. However, the proposed scenario relied on a questionable inversion of limits and was, thus, not completely satisfactory.

An attempt to circumvent that problem was proposed in Ref. [19] in an extension of the procedure of Ref. [16] to the class of nonlinear Curci-Ferrari-Delbourgo-Jarvis gauges. It was shown there that a gluon mass is indeed dynamically generated due to collective effects at large distances. Unfortunately, the latter tends to zero in the Landau gauge limit. In the present paper, we come back to a simpler setup by considering a slight deformation of the original proposal of Ref. [16], directly in the Landau gauge. We show, first, that there exists some

values of the gauge-fixing parameters for which the symmetry of the NL$\sigma$ sector is radiatively restored and, second, that, whenever this happens, this results in a nonzero gluon mass at tree level.

## 2 Gauge fixing

In this Section, we describe our gauge-fixing procedure and its formulation in terms of a local field theory with auxiliary fields. We concentrate on the Landau gauge, for which lots of information were obtained by lattice simulations. In practice, when a gauge configuration $A_\mu = A_\mu^a t^a$ (with $t^a$ the generators of the group in the fundamental representation) has been selected, one has to find a gauge transformation $U$ such that

$$\partial_\mu A_\mu^U = 0, \tag{1}$$

where the gauge transformation reads $A_\mu^U = U A_\mu U^\dagger + \frac{i}{g} U \partial_\mu U^\dagger$. The gauge can be equivalently fixed by imposing that $U$ extremizes

$$f[A, U] \equiv \int d^d x \, \text{tr} \left( A_\mu^U \right)^2, \tag{2}$$

at fixed $A_\mu$. As first stressed by Gribov [13], there exists in general many solutions $U_i$, called Gribov copies, to this problem. To characterize unambiguously the gauge-fixing procedure, it is necessary to supplement the condition given in Eq. (1) with a rule describing how to deal with these Gribov copies. The general strategy put forward in Ref. [16] is to sum over the different copies, with a nonuniform weight function $\mathcal{P}[A, U]$. The gauge-fixing procedure applied to some operator $\mathcal{O}$ is then:

$$\langle\langle \mathcal{O}[A] \rangle\rangle \equiv \frac{\sum_i \mathcal{O}[A^{U_i}] \mathcal{P}[A, U_i]}{\sum_i \mathcal{P}[A, U_i]}, \tag{3}$$

where the sums run over all Gribov copies. As it should be for a *bona fide* gauge fixing, a gauge-invariant operator is not modified by this procedure, $\langle\langle \mathcal{O}_{\text{inv}} \rangle\rangle = \mathcal{O}_{\text{inv}}$, thanks to the denominator appearing in Eq. (3). This implies in particular that the average of a gauge-invariant observable does not depend on the particular choice of weight function $\mathcal{P}$.

In this article, we use

$$\mathcal{P}[A, U] \equiv \frac{\text{Det}(\mathcal{F}[A, U] + \zeta \mathbb{1})}{|\text{Det}(\mathcal{F}[A, U])|} e^{-\beta f[A, U]}, \tag{4}$$

where $\mathcal{F}[A, U]$ is the Hessian of the functional (2) on the group manifold at fixed $A$, that is, the FP operator in the Landau gauge $\mathcal{F}[A, U] = \hat{\mathcal{F}}[A^U]$, with $\hat{\mathcal{F}}^{ab}[A; x, y] \equiv -\partial_\mu [\partial_\mu \delta^{ab} + g f^{acb} A_\mu^c(x)] \delta^{(d)}(x - y)$. Here, $\beta$ and $\zeta$ are two gauge-fixing parameters of mass dimension 2. For $\zeta = 0$, the ratio of functional determinants on the right-hand side reduces to the sign of the determinant of the FP operator and the weight (4) identifies with the one proposed in Ref. [16]. In this case, the resulting local field theory (see below) is of topological nature with the important consequence that the corresponding auxiliary fields give no loop contributions. A nonzero $\zeta$ relaxes those topological constraints and allows for a richer structure, in particular, the possibility of radiative symmetry restoration.

It is also interesting to note that the gauge-fixing proposed here, although very different in spirit, shares some qualitative similarities with the renown (refined) Gribov-Zwanziger (GZ) approach [13, 14, 20, 21]. In the latter, the integral over gauge field configurations is

restricted to the first Gribov region—where the FP operator is positive definite—which eventually strongly favors configurations near the first Gribov horizon—where the FP operator has its first zero eigenvalue. The present approach involves copies over all Gribov regions but the weight in (3) can be engineered so as to favor similar configurations as in the GZ approach. With the choice (4), for $\zeta \neq 0$, the numerator favors the copies near the Gribov horizons[1] whereas the exponential factor suppresses the contribution from higher-than-the-first Gribov regions. It may be tempting to try to connect the various gauge-fixing parameters of mass dimension 2 involved in both the GZ and the present approaches but these are really different gauge-fixings and such a precise connection is certainly nontrivial.

Rewriting the gauge-fixing procedure proposed here in terms of a continuum field theory is standard [16] and involves introducing auxiliary fields. The numerator of Eq. (3) can be rewritten as

$$\sum_i \mathcal{O}[A^{U_i}]\mathcal{P}[U^i] = \int \mathcal{D}U\mathcal{D}c\mathcal{D}\bar{c}\mathcal{D}h\, \mathcal{O}[A^U]e^{-S_{\mathrm{gf}}[A^U,c,\bar{c},h]}, \tag{5}$$

where $U$ is a matrix field living in the gauge group (the integral implicitly involves the corresponding Haar measure), $c$, $\bar{c}$, and $h$ are, respectively, a pair of Grassmann (ghost and antighost) fields and the Nakanishi-Lautrup field which ensures the Landau gauge condition, all taking values in the Lie algebra, and[2]

$$S_{\mathrm{gf}}[A,c,\bar{c},h] = \int d^d x \left[ \partial_\mu \bar{c}^a (\partial_\mu c^a + g f^{abc} A_\mu^b c^c) + \zeta \bar{c}^a c^a + ih^a \partial_\mu A_\mu^a + \frac{\beta}{2}(A_\mu^a)^2 \right]. \tag{6}$$

These auxiliary fields can be conveniently merged into a superfield [16]

$$\mathcal{V}(x,\theta,\bar{\theta}) \equiv e^{ig(\bar{\theta}c + \bar{c}\theta + \bar{\theta}\theta\tilde{h})}U(x), \tag{7}$$

where $\theta$ and $\bar{\theta}$ are (anticommuting) Grassmann variables. Here, the algebra-valued fields which appear without color index are implicitly contracted with the generators of the algebra, e.g., $c = c^a t^a$, and $\tilde{h} = ih - i\frac{g}{2}\{\bar{c},c\}$. The superfield $\mathcal{V}$ takes values in the gauge group and, for SU($N$), satisfies $\mathcal{V}^\dagger \mathcal{V} = \mathbb{1}$. The gauge-fixing action simply rewrites

$$S_{\mathrm{gf}}[A^U,c,\bar{c},h] = \widetilde{S}_{\mathrm{gf}}[A,\mathcal{V}] = \frac{1}{g^2}\int_x \int_{\underline{\theta}} \mathrm{tr}\left[(\mathcal{D}_\mu \mathcal{V})^\dagger (\mathcal{D}_\mu \mathcal{V}) + 2\zeta\theta\bar{\theta}\partial_{\bar{\theta}}\mathcal{V}^\dagger \partial_\theta \mathcal{V}\right], \tag{8}$$

where $\int_x = \int d^d x$, $\int_{\underline{\theta}} = \int d\theta d\bar{\theta}(\beta\bar{\theta}\theta - 1)$ and the covariant derivative is defined as $\mathcal{D}_\mu \mathcal{V} = \partial_\mu \mathcal{V} + ig\mathcal{V}A_\mu$. Our conventions are such that $\int_{\underline{\theta}} 1 = \beta$ and $\int_{\underline{\theta}} \bar{\theta}\theta = -1$. Note the invariance of the action (8) under the gauge transformation

$$\widetilde{S}_{\mathrm{gf}}[A,\mathcal{V}] = \widetilde{S}_{\mathrm{gf}}[A^U,\mathcal{V}U^{-1}], \tag{9}$$

where $U(x)$ is an arbitrary element of the gauge group.

We treat the denominator in Eq. (3) with the replica trick, summarized by the identity $1/a = \lim_{n\to 0} a^{n-1}$. We introduce $n-1$ copies of the integration variable $\mathcal{V}$ defined above

---

[1]The weight (4) is singular for configurations on Gribov horizons, where the FP determinant vanishes. We see no obvious sign of such singularity in the continuum formulation presented below and we shall simply assume that it does not lead to any pathology under the path integral. It could be that such singular configurations are, in fact, of zero measure (note, however, that this would question the qualitative analogy with the dominant configurations in the GZ approach mentioned in the text). Another possibility is that for those configurations, the singularity in $\mathcal{P}[A,U]$ cancels out between the numerator and the denominator in Eq. (3). Let us also stress that the choice (4) leads to a renormalizable continuum theory, as discussed below. The possible singularities due to the denominator in (4) could be smoothened by nonrenormalizable terms which are expected to play no role for the large distance physics.

[2]The FP action corresponds to the case $\beta = \zeta = 0$, that is, to a uniform weight in Eq. (4).

which, together with the numerator, give $n$ replicas $\mathcal{V}_{k=1,\dots,n}$ of the supersymmetric field and we have to take the limit $n \to 0$ at the end of any calculation. Once the gauge-fixing procedure is realized, we average over the gauge field configurations, with the Yang-Mills weight:

$$\langle \mathcal{O}[A] \rangle = \frac{\int \mathcal{D}A\, e^{-S_{\mathrm{YM}}[A]} \langle\langle \mathcal{O}[A] \rangle\rangle}{\int \mathcal{D}A\, e^{-S_{\mathrm{YM}}[A]}} = \lim_{n \to 0} \frac{\int \mathcal{D}A \left( \prod_{k=1}^{n} \mathcal{D}\mathcal{V}_k \right) e^{-S_{\mathrm{YM}}[A] - \widetilde{S}_{\mathrm{gf}}[A, \mathcal{V}_k]} \mathcal{O}[A^{U_1}]}{\int \mathcal{D}A \left( \prod_{k=1}^{n} \mathcal{D}\mathcal{V}_k \right) e^{-S_{\mathrm{YM}}[A] - \widetilde{S}_{\mathrm{gf}}[A, \mathcal{V}_k]}}, \quad (10)$$

where the second expression exploits the replica trick. At this level, all replicas are trivially equivalent. We can now factorize the volume of the gauge group by absorbing the dependence of the integrand in one of the replicated matrix fields, say, $U_1$. Changing the integration variables to $A_\mu \to A_\mu^{U_1}$, $\mathcal{V}_k \to \mathcal{V}_k U_1^{-1}$, it is straightforward to check that the functional integrands in Eq. (10) are now independent of $U_1$ and one can factor out the volume of the gauge group $\int \mathcal{D}U_1$. As usual, this step is what allows for a well-defined gluon propagator. The choice of the replica $k = 1$ is, of course, arbitrary here and, as explained in Ref. [16], the permutation symmetry between replicas remains intact. We end up with the following gauge-fixed action

$$S = S_{\mathrm{YM}}[A] + S_{\mathrm{gf}}[A, c, \bar{c}, h] + \sum_{k=2}^{n} \widetilde{S}_{\mathrm{gf}}[A, \mathcal{V}_k], \quad (11)$$

which involves the gluon field $A$, a pair of ghost/antighost fields $c$ and $\bar{c}$, a Nakanishi-Lautrup field $h$, which all live in the Lie algebra of the group, and $n-1$ supersymmetric fields $\mathcal{V}_k$, which take values in the gauge group.

It was shown in Ref. [16] that, in the case $\zeta = 0$, the (super)symmetries of the replica sector guarantee that all closed loops of the replica fields vanish. Also, the gauge-fixed action (11) was proven to be perturbatively renormalizable in $d = 4$ in that case. The case $\zeta \neq 0$ simply adds operators of mass dimension 2, which should not spoil renormalizability, although we leave the study of that precise point for later. This, however, spoils the supersymmetries mentioned above and, as we shall see explicitly below, the replica sector now yields nontrivial loop contributions. The latter are crucial for the possibility of radiatively induced symmetry restoration mentioned in the Introduction.

## 3 Symmetry restoration

The auxiliary superfields $\mathcal{V}_k$ introduced above take values in the SU($N$) gauge group and, as already mentioned, resemble closely the constrained fields of a NL$\sigma$ model in 2 dimensions, which are known to display the phenomenon of symmetry restoration. To make the discussion simpler we focus in the remainder of the article on the SU(2) gauge group. The generators of the algebra in the fundamental representation are given by $t^a = \sigma^a/2$, with $\sigma^{a=1,2,3}$ the Pauli matrices, and the group elements are conveniently written in terms of unit four-component fields $N^{\alpha=0,1,2,3}$, as $\mathcal{V}_k = N_k^\alpha \Sigma^\alpha$, where $\Sigma^\alpha = \{\mathbb{1}, i\sigma^a\}$. Here and below, latin indices run from 1 to 3 and the greek indices at the beginning of the alphabet run from 0 to 3 and are associated with the group structure. The gauge-fixing action (8) in the replica sector reads

$$\widetilde{S}_{\mathrm{gf}}[A, \mathcal{V}_k] = \int_x \frac{\beta}{2} (A_\mu^a)^2 + \frac{2}{g^2} \int_x \int_{\underline{\theta}} \left\{ (\partial_\mu N_k^\alpha)^2 + g f^{a\alpha\beta} A_\mu^a N_k^\alpha \partial_\mu N_k^\beta + 2\zeta \theta \bar{\theta} \partial_{\bar{\theta}} N_k^\alpha \partial_\theta N_k^\alpha \right\}, \quad (12)$$

where we have used $\mathcal{V}^\dagger \mathcal{V} = N^2 = N^\alpha N^\alpha = 1$ in the first term on the right-hand side and where we introduced the tensor $f^{a\alpha\beta} = -\frac{i}{4}\mathrm{tr}\{\sigma^a[(\Sigma^\alpha)^\dagger \Sigma^\beta - (\Sigma^\beta)^\dagger \Sigma^\alpha]\}$. The latter is antisymmetric in its last two indices and is fully characterized by $f^{a0b} = \delta^{ab}$ and $f^{abc} = \epsilon^{abc}$. We mention the identity $f^{a\alpha\beta} f^{b\alpha\beta} = 4\delta^{ab}$.

Let us pause a moment to describe qualitatively the scenario we want to explore here. The action (12) involves a (supersymmetric) O(4)/O(3) NL$\sigma$ model coupled to the gluon field. Qualitatively, this corresponds to the broken phase of a (supersymmetric) Higgs model, hence the appearance of a mass term for the gauge field. All replicas $k = 2, \ldots, n$ in Eq. (11) contribute the same to the gluon mass term. Together with the replica $k = 1$, included in the second term on the right-hand side of Eq. (11), the total tree-level gluon square mass is $\beta + (n-1)\beta = n\beta$, which vanishes in the limit $n \to 0$. However, another scenario is possible. From the usual Higgs model, we expect the contribution to the mass of the gauge field from the replica sector to vanish in the symmetric phase. In that case, the gluon square mass only receives a contribution $\beta$ from the replica $k = 1$, which is nonzero in the limit $n \to 0$.

To investigate the possibility of symmetry restoration, we follow the approach of Refs. [22, 23] and relax the constraint of unit length on the (super)fields $N_k^\alpha$ by introducing Lagrange multiplier (super)fields $\chi_k$ for the replica $k = 2, \ldots, n$ in Eq. (11) as

$$\widetilde{S}_{\text{gf}}[A, \mathcal{V}_k] \to \widetilde{S}_{\text{gf}}[A, N_k, \chi_k] = \widetilde{S}_{\text{gf}}[A, \mathcal{V}_k] + \frac{2}{g^2} \int_x \int_{\underline{\theta}} i\chi_k \left(N_k^2 - 1\right). \tag{13}$$

Integrating over the real superfields $\chi_k$ imposes the hard constraints $N_k^2 = 1$ under the path integral. The main purpose of using the action (13) is that one can then perform exactly the Gaussian integration over the (unconstrained) fields $N_k^\alpha$ and study the effects of the corresponding fluctuations on the other fields. In practice, we shall thus integrate out the $N_k^\alpha$ fluctuations exactly, while treating the other fields at the classical (tree-level) order, as described in the Appendix A.

We then study the equations of motion for the (real) averaged fields $\hat{\chi}_k \equiv \langle i\chi_k \rangle$ and $\hat{N}_k^\alpha \equiv \langle N_k^\alpha \rangle$. For the present purposes, it is sufficient to consider field configurations independent of spacetime and Grassmann coordinates. We also choose them independent of the replica index, $\hat{\chi}_k(x, \theta, \bar{\theta}) = \hat{\chi}$ and $\hat{N}_k^\alpha(x, \theta, \bar{\theta}) = \hat{N}^\alpha$, assuming that the permutation symmetry between the replicas $k = 2, \ldots, n$ is left unbroken. At the order of approximation considered here, the relevant equations of motion, $\langle \delta S / \delta \chi_k \rangle = \langle \delta S / \delta N_k^\alpha \rangle = 0$, read

$$\int_{\underline{\theta}} \left\langle N_k^2 - 1 \right\rangle = 0 \quad \text{and} \quad \hat{\chi} \hat{N}^\alpha = 0. \tag{14}$$

The second equation only receives a purely classical contribution whereas the first one involves both a classical contribution from the average $\hat{N}^\alpha$ and a loop contribution given by the integral of the $N_k^\alpha$ propagator over momentum, over the Grassmann coordinates $\underline{\theta}$, and summed over the index $\alpha$. At the considered order of approximation, this propagator is given by its tree-level expression, where both $\hat{\chi}$ and $\hat{\chi} + \zeta$ appear as square masses of propagators and are therefore restricted to be nonnegative. This yields the equation

$$\frac{\beta}{2\bar{g}^2} \left(\hat{N}^2 - 1\right) + T_{\hat{\chi}} - T_{\hat{\chi} + \zeta} = 0, \tag{15}$$

where we have introduced the dimensionless coupling $\bar{g} = g\mu^{-\epsilon}$, with $d = 4 - 2\epsilon$ and $\mu$ is an arbitrary mass scale. In dimensional regularization, the tadpole loop integrals are given by

$$T_{m^2} \equiv \mu^{2\epsilon} \int \frac{d^d p}{(2\pi)^d} \frac{1}{p^2 + m^2} = -\frac{m^2}{16\pi^2} \left[\frac{1}{\epsilon} + 1 + \ln \frac{\bar{\mu}^2}{m^2} + O(\epsilon)\right], \tag{16}$$

where $\bar{\mu}^2 = 4\pi e^{-\gamma}\mu^2$, with $\gamma$ the Euler constant. As expected, the loop contribution in Eq. (15) vanishes at $\zeta = 0$. This implies that the loop-divergence is proportional to $\zeta$ and thus only logarithmic, a manifestation of the dimensional reduction mentioned above.

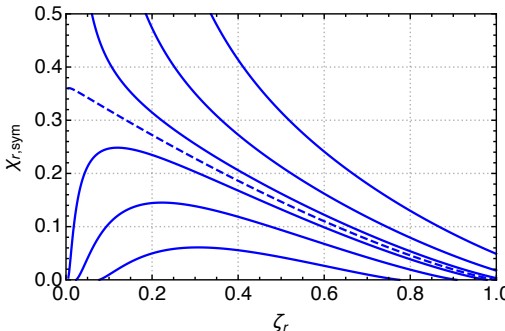

Figure 1: The classical field $\hat{\chi}_{r,\text{sym}}$ as a function of the gauge parameter $\zeta_r$ for increasing (top to bottom) values of $\beta_r/\bar{g}_r^2$ (in units of $\bar{\mu}^2$). The dashed curve is the case $\beta_r/\bar{g}_r^2 = 0$. For positive values of this parameter (curves below the dashed one), solutions of the gap equation (22) with $\hat{\chi}_{r,\text{sym}} \geq 0$ only exist in a range of parameters limited by the inequalities (23).

The system of equations (14) and (15) has two solutions, with either $\hat{\chi} = 0$ or $\hat{N}^\alpha = 0$. The one with $\hat{\chi} = 0$ corresponds to the phase of broken O(4) symmetry, with the constraint $\hat{N}_{\text{brok}}^2 = \text{const}$. As already mentioned, $\hat{\chi}$ plays the role of a square mass for the bosonic components of the superfield $N_k^\alpha$, which are nothing but the Goldstone modes in that phase. We have, from Eq. (15),

$$\hat{N}_{\text{brok}}^2 = 1 + \frac{2\bar{g}^2}{\beta} T_\zeta. \tag{17}$$

For $\zeta = 0$, the (superfield) loop contribution in Eq. (15) vanishes, leaving the broken phase with the tree-level constraint $\hat{N}_{\text{brok}}^2 = 1$ as the only solution. In constrast, the case $\zeta > 0$ allows for the other solution to Eq. (14), with $\hat{N}^\alpha = 0$. In the absence of the coupling to the gauge field this would correspond to a restored O(4) symmetry. In the following, we refer to this solution as the symmetric phase. It is characterized by massive modes with square mass $\hat{\chi} = \hat{\chi}_{\text{sym}} > 0$, solution of the gap equation

$$\frac{\beta}{2\bar{g}^2} = T_{\hat{\chi}_{\text{sym}}} - T_{\hat{\chi}_{\text{sym}}+\zeta} = \frac{1}{16\pi^2}\left[\frac{\zeta}{\epsilon} + \zeta + (\hat{\chi}_{\text{sym}} + \zeta)\ln\frac{\bar{\mu}^2}{\hat{\chi}_{\text{sym}}+\zeta} - \hat{\chi}_{\text{sym}}\ln\frac{\bar{\mu}^2}{\hat{\chi}_{\text{sym}}}\right]. \tag{18}$$

The above equations are UV divergent and require renormalization. We introduce the renormalized fields and parameters as

$$N_k^\alpha = \sqrt{Z_N}N_{k,r}^\alpha, \quad \chi = \sqrt{Z_\chi}\chi_r, \quad \beta = Z_\beta\beta_r, \quad \zeta = Z_\zeta\zeta_r, \quad \bar{g} = Z_g\bar{g}_r. \tag{19}$$

Note that the first (classical) term on the left-hand side of Eq. (15) receives an overall factor $\sqrt{Z_\chi}$. Also, at the present order of approximation, we can set all renormalisation factors $Z \to 1$ in the tadpole (one-loop) integrals. We eliminate the UV divergence in Eq. (15) with the choices

$$\sqrt{Z_\chi}Z_\beta Z_g^{-2} = Z_N^{-1} = 1 + \frac{\bar{g}_r^2}{8\pi^2}\frac{\zeta_r}{\beta_r}\left(\frac{1}{\epsilon} + 1\right). \tag{20}$$

The broken phase solution (17) rewrites

$$\hat{N}_{r,\text{brok}}^2 = 1 - \frac{\bar{g}_r^2}{16\pi^2}\frac{\zeta_r}{\beta_r}\ln\frac{\bar{\mu}^2}{\zeta_r}, \tag{21}$$

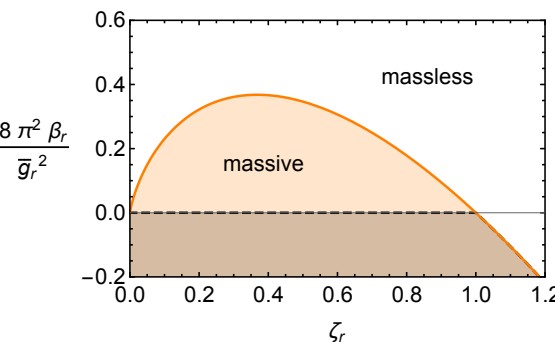

Figure 2: The phase diagram of the theory in terms of the parameters $\beta_r/\bar{g}_r^2$ and $\zeta_r$ (in units of $\bar{\mu}^2$). The orange line separates the phase of broken symmetry (white region), where there is no generation of a gluon mass at tree level, from that of restored symmetry (shaded resion), with $\hat{N}^\alpha = 0$, where a nonzero gluon mass is generated at tree level. The transition between the two phases is continuous. In the massive phase, the renormalized tree-level square gluon mass is $\beta_r$. The shaded region with $\beta_r < 0$ is thus excluded in a perturbative setting.

whereas the gap equation (18) in the symmetric phase becomes

$$\frac{8\pi^2 \beta_r}{\bar{g}_r^2} = (\hat{\chi}_{r,\text{sym}} + \zeta_r)\ln\frac{\bar{\mu}^2}{\hat{\chi}_{r,\text{sym}} + \zeta_r} - \hat{\chi}_{r,\text{sym}}\ln\frac{\bar{\mu}^2}{\hat{\chi}_{r,\text{sym}}}. \tag{22}$$

The right-hand side is a monotonously decreasing function of $\hat{\chi}_{r,\text{sym}}$ so there exists a unique solution if and only if

$$\frac{8\pi^2 \beta_r}{\bar{g}_r^2} \leq \zeta_r \ln\frac{\bar{\mu}^2}{\zeta_r} \leq \frac{\bar{\mu}^2}{e}, \tag{23}$$

where the second inequality is a bound for all values of the parameter $\zeta_r$. The behavior of $\hat{\chi}_{r,\text{sym}}$ as a function of $\zeta_r$ is presented in Fig. 1. For completeness, we show results for positive and negative values of $\beta_r/\bar{g}_r^2$. We mention though that, when a solution $\hat{\chi}_{r,\text{sym}} > 0$ exists, the parameter $\beta_r$ plays the role of the renormalized tree-level gluon mass (see below) and is thus restricted to be nonnegative in a perturbative setting. The broken-symmetry *vs.* symmetric phases discussed here only coexist when Eq. (21) yields $\hat{N}^2_{r,\text{brok}} = 0$ or, equivalently, when Eq. (22) is satisfied at $\hat{\chi}_{r,\text{sym}} = 0$. This happens for values of the parameters which saturate the first inequality in Eq. (23). The phase diagram of the theory is shown in Fig. 2.

## 4 Mass generation

Having established the phenomenon of radiative symmetry restoration, we now check the expectation that the contribution to the gluon mass from the replica field indeed vanishes in this phase. To this end, we consider the effective action of the theory $\Gamma[\hat{A}, \hat{\chi}, \hat{N}]$, where $\hat{A} = \langle A \rangle$. Integrating out the superfields $N_k^\alpha$ exactly and treating the other fields at tree level, we get, in terms of bare quantities (see Appendix A),

$$\Gamma[\hat{A}, \hat{\chi}, \hat{N}] = S_{\text{YM}}[\hat{A}] + \int_x \frac{\beta}{2}(\hat{A}_\mu^a)^2 + (n-1)\int_x \left\{\frac{\beta}{2}(\hat{A}_\mu^a)^2 + \frac{2\beta}{g^2}\hat{\chi}(\hat{N}^2 - 1)\right\}$$
$$+ (n-1)\left\{\text{Tr}\ln\left[-\partial^2 + \hat{\chi} + ig\hat{A}_\mu \partial_\mu\right] - \text{Tr}\ln\left[-\partial^2 + \hat{\chi} + \zeta + ig\hat{A}_\mu \partial_\mu\right]\right\} + \text{loops}, \tag{24}$$

where we have introduced the matrix field $(\hat{A}_\mu)^{\alpha\beta} = -i\hat{A}_\mu^a f^{a\alpha\beta}$ and where the neglected contributions (loops) involve the fluctuations of the fields $A$, $c$, $\bar{c}$, and $\chi_k$. Here, the first line is simply the classical action and the trace-log terms in the second line are the (loop) contributions from the $N_k^\alpha$ fluctuations. As expected, the loop contribution from the replica sector identically vanishes for $\zeta = 0$. The part quadratic in $\hat{A}_\mu^a$ gives the inverse gluon propagator of the theory. Writing the latter in momentum space as

$$\left.\frac{\delta\Gamma}{\delta\hat{A}_\mu^a(q)\delta\hat{A}_\nu^b(-q)}\right|_{\hat{A}=0} = \delta^{ab}\left[q^2\delta_{\mu\nu} - q_\mu q_\nu + \beta\delta_{\mu\nu} + \Pi_{\mu\nu}^{\text{rep}}(q)\right], \tag{25}$$

where $\Pi_{\mu\nu}^{\text{rep}}$ is the contribution from the loop of replica fields $N_k^\alpha$, as represented in Fig. 3. At the present approximation order, a straightforward calculation gives

$$\Pi_{\mu\nu}^{\text{rep}}(q) = (n-1)\beta\delta_{\mu\nu} + (n-1)\bar{g}^2\left[\mathcal{I}_{\mu\nu}^{\hat{\chi}+\zeta}(q) - \mathcal{I}_{\mu\nu}^{\hat{\chi}}(q)\right], \tag{26}$$

with the integral

$$\mathcal{I}_{\mu\nu}^{m^2}(q) = \mu^{2\epsilon}\int\frac{d^dp}{(2\pi)^d}\frac{(2p-q)_\mu(2p-q)_\nu}{[p^2+m^2][(p-q)^2+m^2]} = 2\delta_{\mu\nu}T_{m^2} + \left(q^2\delta_{\mu\nu} - q_\mu q_\nu\right)F_{m^2}(q^2). \tag{27}$$

In the second equality, we have defined

$$F_{m^2}(q^2) = -\frac{1}{48\pi^2}\left[\frac{1}{\epsilon} + \frac{8}{3} + \ln\frac{\bar{\mu}^2}{m^2} + \frac{8m^2}{q^2} - 2\left(1+\frac{4m^2}{q^2}\right)^{3/2}\ln\left(\frac{q}{2m} + \sqrt{1+\frac{q^2}{4m^2}}\right)\right]. \tag{28}$$

We observe that the only UV divergence in Eq. (26) is momentum independent, another consequence of the effective dimensional reduction mentioned above. This divergence is the one of the gap equation and can be absorbed in the same way. We rewrite Eq. (26) as

$$\Pi_{\mu\nu}^{\text{rep}}(q) = (n-1)\delta_{\mu\nu}\left[\beta - 2\bar{g}^2\left(T_{\hat{\chi}} - T_{\hat{\chi}+\zeta}\right)\right] + \left(q^2\delta_{\mu\nu} - q_\mu q_\nu\right)\pi(q^2), \tag{29}$$

with

$$\begin{aligned}\pi(q^2) &= (n-1)\bar{g}^2\left[F_{\hat{\chi}+\zeta}(q^2) - F_{\hat{\chi}}(q^2)\right]\\&= -(n-1)\frac{g^2}{48\pi^2}\left[\ln\frac{\hat{\chi}}{\hat{\chi}+\zeta} + \frac{8\zeta}{q^2} + \mathcal{F}\left(\frac{q^2}{4\hat{\chi}}\right) - \mathcal{F}\left(\frac{q^2}{4(\hat{\chi}+\zeta)}\right)\right],\end{aligned} \tag{30}$$

where

$$\mathcal{F}(x) = 2\left(1+\frac{1}{x}\right)^{3/2}\ln\left(\sqrt{x} + \sqrt{1+x}\right). \tag{31}$$

The function $\pi(q^2)$ is regular at $q^2 = 0$ and the effective gluon mass, defined from the vertex function (25) at $q = 0$, thus reads

$$m_g^2 = \beta + (n-1)\left[\beta - 2\bar{g}^2\left(T_{\hat{\chi}} - T_{\hat{\chi}+\zeta}\right)\right] = \beta\left[1 + (n-1)\hat{N}^2\right], \tag{32}$$

where we have used the gap equation (15) in the second equality. We thus see that, in the symmetric phase, the loop and tree-level contributions to the gluon screening mass exactly cancel each other in the replica sector (even before the limit $n \to 0$), so that $m_g^2 = \beta$. In that case, it is necessary to take the corresponding contribution of the replica loops into account at tree level for the gluon propagator (the momentum dependent part is $\mathcal{O}(g^2)$ and is to be treated as a loop contribution, at the same level as gluon and ghost loops). In the broken phase, instead, we have $m_g^2 = n\beta + \mathcal{O}(g^2)$ so the replica loops have to be completely considered as

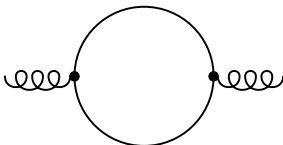

Figure 3: The loop contribution from the $N_k^\alpha$ superfields to the gluon two-point vertex function.

loop effects and the effective tree-level gluon mass is $m_g^2 = n\beta \to 0$. The two, massive *vs.* massless phases of the theory are summarized in Fig. (2).

It is worth emphasizing that the phenomenon of mass generation described here differs somewhat from the more usual cases of, say, the fermion masses in the electroweak theory or the constituent quark masses in QCD. In both cases, the generated mass is controlled by a dynamically determined quantity, typically a condensate. In the present case, the gluon mass is controlled by a parameter of the Lagrangian and it may seem that it can take any value. We stress, however, that the mass is nonzero only for specific values of the parameters— corresponding to the massive phase—and its value is not completely arbitrary, being bounded, at tree-level, as $0 < m_g^2 < \beta$; see Eq. (32). Also, the fact that the generated gluon mass is directly given by the gauge-fixing parameter $\beta$ is of no particular worry since it is a gauge-dependent quantity. For instance, the gluon screening mass observed in lattice calculations depends on the particular way the Gribov problem is handled, that is, *in fine*, on the choice of gauge [12, 15].

## 5 Conclusions

To summarize, we have studied the relation between the Gribov ambiguity issue and the generation of a gluon (screening) mass in Yang-Mills theories in the Landau gauge. We have used a gauge-fixing procedure that mimics some aspects of the minimal Landau gauge, widely used in lattice simulations, where a random Gribov copy is selected in the first Gribov region. The main advantage on the latter is that our procedure can be formulated in terms of a local gauge-fixed action. It is a slight generalization of the proposal put forward in Ref. [16], where Gribov copies are averaged over with a weight expected to typically favor the first Gribov region. The extension proposed here adds some weights on copies near the first Gribov horizon, a feature shared with the GZ quantization procedure.[3] Overall, this involves two gauge-fixing parameters, $\beta$ and $\zeta$, of mass dimension 2.

The corresponding gauge-fixed action contains a set of $n$ supersymmetric NL$\sigma$ models which effectively behave as two-dimensional. Due to their coupling to the gauge field, these all contribute a tree-level gluon square mass $\beta$, much alike the Higgs phenomenon, giving a total square mass $n\beta$ which vanishes in the physically relevant limit $n \to 0$. One of these

---

[3]We mention that an asset of the present construction is that it treats consistently all Gribov copies as opposed to the GZ approach which, by restricting to the first Gribov region, deals only with the so-called infinitesimal copies. Our approach, however, requires that the Gribov copies correspond to extrema of some functional. This is the case of the Landau gauge and, for example, of the class of Curci-Ferrari-Delbourgo-Jarvis gauges [34, 35], which have been studied in terms of a similar averaging procedure in [36]. Unfortunately, this does not apply to linear gauges which are not easily expressed as an extremization procedure but for which interesting progress has been made both in lattice simulations [37, 38] and in the GZ approach [39].

replica fields is singled out to factor out the volume of the gauge group as required for properly defining a perturbative gluon propagator. The remaining replica fields exhibit a phenomenology reminiscent of the standard NL$\sigma$ model in two dimensions, namely, a phase of radiatively restored symmetry. In the symmetric phase, the underlying gauge symmetry guarantees that the contribution from the NL$\sigma$ sectors to the tree-level gluon mass vanishes. The permutation symmetry between the replica is spontaneously broken and only the replica singled out to factor out the volume of the gauge group now contributes a tree-level gluon square mass $\beta$, which remains in the limit $n \to 0$.

Of course, one must keep in mind that the gauge-fixing procedure proposed here is not the same as the minimal Landau gauge used in most lattice simulations. But it provides an explicit example where a nonzero gluon mass term is generated from the gauge-fixing procedure, in particular, from the treatment of the Gribov copies. The resulting gauge-fixed theory resembles the perturbative Curci-Ferrari model [24] which has been recently applied with success to describe a variety of lattice results both in the vacuum and at finite temperature [25, 27–29]. Eventually, on top of the gluon mass term present in that model, here, with square mass $\beta$, our gauge-fixed action features a ghost mass term with square mass $\zeta$, as well as a set of massive fields with square masses $\hat\chi_{\text{sym}}$ and $\hat\chi_{\text{sym}} + \zeta$, self-consistently determined through the gap equation (22). We stress that the presence of massive ghosts is not directly in conflict with lattice results, where the ghost propagator is seen to diverge as that of a massless field at infrared momenta. This is because, what is actually measured on the lattice is not directly the ghost propagator but the averaged inverse FP operator $\langle \hat{\mathcal{F}}^{-1}[A] \rangle$, see Eq. (4). The latter coincides with the ghost propagator in the FP implementation of the Landau gauge but that is not the general case, as the present proposal shows.

We shall discuss these aspects further in a future work. For instance, it is of interest to compute the propagators of the various fields in the present gauge fixing at one-loop order and compare with the results of the Curci-Ferrari model and with lattice simulations. In particular, the massive ghosts will play a role through loop contributions. Other interesting aspects to be investigated are the renormalizability of the present gauge-fixed theory and its renormalization group trajectories.

## Acknowledgements

This study was financed in part by the Coordenação de Aperfeiçoamento de Pessoal de Nível Superior - Brasil (Capes) - Finance Code 001. We also acknowledge support from the program ECOS Sud U17E01.

## A   Approximation scheme

We detail the mixed approximation scheme used in this work, where we have treated the fields $A$, $c$, $\bar c$, $h$, and $\chi_k$ at tree-level while exactly integrating out the fields $N_k^\alpha$. As explained in the main text, the aim is to investigate the possibility that the corresponding fluctuations yield a phase with $\langle N_k^\alpha \rangle = 0$, which we refer to as the symmetric phase. When this happens, some loop contributions to the total effective action effectively contribute at tree level and must, therefore, be systematically included.

To simplify matters, we consider a simpler model with two (nonsupersymmetric) fields $\varphi$, to be treated at tree-level, and $n$, to be integrated out and which only appears quadratically in the microscopic action. The derivation presented here easily generalizes to the model

developed in the main text. The generating functional for field correlators $W$ is defined as

$$e^{W[J,j]} = \int \mathcal{D}\varphi \mathcal{D}n \, e^{-S[\varphi]-\frac{1}{2}n\cdot G^{-1}[\varphi]\cdot n + j\cdot n + J\cdot \varphi} \tag{33}$$

$$\propto \int \mathcal{D}\varphi \, e^{-S[\varphi]-\frac{1}{2}\mathrm{TrLn}G^{-1}[\varphi]+\frac{1}{2}j\cdot G[\varphi]\cdot j + J\cdot \varphi}, \tag{34}$$

where the action $S[\varphi]$ and the operator $G^{-1}[\varphi]$ are *a priori* arbitrary. In the second line, we have explicitly performed the Gaussian integration of the field $n$, thus, treating the corresponding loops exactly. Keeping the source $j \neq 0$ at this level is essential to be able to describe $n$ field correlators, in particular, the possibility of a nonzero one-point function $\langle n \rangle$.

At tree level, the functional integral (34) is given by the saddle-point approximation, that is, up to a field-independent contribution,

$$W[J,j] = -S[\phi] - \frac{1}{2}\mathrm{TrLn}G^{-1}[\phi] + \frac{1}{2}j\cdot G[\phi]\cdot j + J\cdot \phi + (\varphi - \mathrm{loops}), \tag{35}$$

where the saddle point $\phi = \phi[J,j]$ is given by

$$\frac{\delta}{\delta\varphi}\left(S[\varphi] + \frac{1}{2}\mathrm{TrLn}G^{-1}[\varphi] - \frac{1}{2}j\cdot G[\varphi]\cdot j - J\cdot \varphi\right)_{\varphi=\phi} = 0 \tag{36}$$

and where the neglected terms involve loop diagrams due to $\varphi$-fluctuations (we recall that the pure $n$-fluctuations are included exactly in the one-loop trace-log term). The corresponding effective action for the average fields $\hat{\phi} = \langle \varphi \rangle$ and $\hat{n} = \langle n \rangle$ is given by the Legendre transform

$$\Gamma[\hat{\varphi},\hat{n}] = -W[J,j] + J\cdot\hat{\varphi} + j\cdot\hat{n}, \tag{37}$$

with

$$\hat{\varphi} = \frac{\delta W}{\delta J} = \phi + (\varphi - \mathrm{loops}) \tag{38}$$

$$\hat{n} = \frac{\delta W}{\delta j} = \langle G[\varphi]\cdot j \rangle = G[\hat{\varphi}]\cdot j + (\varphi - \mathrm{loops}). \tag{39}$$

One obtains

$$\Gamma[\hat{\varphi},\hat{n}] = S[\hat{\varphi}] + \frac{1}{2}\hat{n}\cdot G^{-1}[\hat{\varphi}]\cdot\hat{n} + \frac{1}{2}\mathrm{TrLn}G^{-1}[\hat{\varphi}] + (\varphi - \mathrm{loops}), \tag{40}$$

where the first two terms on the right-hand side simply correspond to the original classical action and the third, trace-log term is the exact contribution from loops of $n$-fields. When applied to the action (12)–(13) with $\varphi \equiv (A, i\chi_k)$ and $n \equiv N_k^\alpha$, this procedure yields the effective action (24).

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
