# Peer review of "Symmetry restoration and the gluon mass in the Landau gauge"

_SciPost Physics, doi:SciPost Phys. 10, 035 (2021)_

## Round 1 · Referee Report · Anonymous (Referee 1) · 2020-10-27

Report
The authors have addressed the issues that I have raised in my previous report and modified the manuscript accordingly.
Their work provides an insightful and potentially powerful way of taking care of Gribov copies in gauges that are obtained through the extremization of a functional. I understand that such a paper opens several important issues to be investigated in the future regarding the consistency of the model and encourage the authors to do so.
As an interesting strength of their method: the averaging over Gribov copies does not impose restrictions on them such as to be generated by infinitesimal gauge transformations.
Therefore, I recommend the paper for publication.
Their work provides an insightful and potentially powerful way of taking care of Gribov copies in gauges that are obtained through the extremization of a functional. I understand that such a paper opens several important issues to be investigated in the future regarding the consistency of the model and encourage the authors to do so.
As an interesting strength of their method: the averaging over Gribov copies does not impose restrictions on them such as to be generated by infinitesimal gauge transformations.
Therefore, I recommend the paper for publication.

---

## Round 1 · Author Response

We thank the referee for the careful reading of our manuscript and for his/her constructive remarks. We answer his/her points and questions below and detail the changes in the manuscript (which appear in red in the revised version).
Referee: It is not clear to me how eq.(4) is defined for configurations that live on the so-called Gribov horizon. Such configurations represent zero modes of the operator F and, therefore, the denominator seems to be ill-defined. The authors could provide a comment on that.
We have added a discussion about this point (footnote 1).
R: The parameter ζ introduces a mass term for the Faddeev-Popov ghosts and the authors claim that it should not be taken as a problem. However, in the standard formulation of the Landau gauge, mass terms for the ghosts are forbidden by the Ward identities. In the standard (Refined) Gribov-Zwanziger construction this remains true. Can the authors comment about that and explain how to reconcile such things? Or am I missing something?
We are not completely sure what the referee means here. The point is that the Ward identities he/she refers to are those of the standard formulation of the Landau gauge, i.e., if we understand correctly, those of the Faddeev-Popov gauge-fixing. But symmetry identities are specific to each particular gauge-fixing and may not apply to others. For instance, a gluon mass term would also be forbidden by the standard Slavnov-Taylor identities. The construction we propose here is a different formulation of the Landau gauge, which comes with its own symmetry identities. In particular, the shift symmetry cbar -> cbar + constant, which forbids a mass term for the ghosts in either the FP or the GW constructions, is simply not a symmetry of the gauge-fixing proposed here.
R: The gluon-mass parameter is associated with the parameter β which is akin to a gauge parameter. This is very confusing to me. Are the authors claiming that the mass parameter that is generated by averaging over Gribov copies is gauge dependent and therefore can be taken to any value? Does it mean that β cannot enter correlation functions of gauge-invariant correlators?
As explained below Eq. (3), gauge-invariant observables are clearly independent of beta (and zeta) by construction. In a systematic expansion scheme (like perturbation theory), we expect this property to be respected order by order.
As for the gluon mass discussed here, it is not a gauge invariant quantity. We see no reason why it should be gauge independent. This, however, does not imply that it can take any value. For instance, as the phase diagram in Fig 2 shows, the value of the generated mass (only in the “massive” phase) is bounded.
Thinking about this point, we have realized that there may be an ambiguity in what one understands by “mass generation”. We have added some comments to clarify these points.
R: Can the authors connect the mass parameter that they obtain to the so-called Gribov parameter which is generated in the standard elimination of infinitesimal Gribov copies in the Landau gauge by the restriction of the path integral to the Gribov region?
No, we do not know how to do such a connection in a precise sense. We only have a qualitative intuition that our procedure favors configurations similar to those which are selected by the GZ restriction, i.e. near the first Gribov horizon. Indeed, the first horizon is enhanced by the denominator in (4) whereas higher Gribov regions are supressed by the exponential weight. It is certainly very hard to make a precise connection between the present proposal and the GZ procedure beyond this qualitative level, simply because these are in fact different gauge-fixing procedures. Relating the dimensionful parameters of these two to one another is certainly nontrivial. We have added a discussion on this point.
R: Below eq.(4), the authors say that F represents the FP operator in the Landau gauge and write F[A,U]=F[A^U] Can they explain what do they mean by this equality?
We simply mean that the function F[A,U] is, in fact, a function of A^U only and not of A and U separately. The revised version makes this clear.
R: It seems that this averaging method does not (strongly) rely on the choice of the Landau gauge and also on the type" of copies, i.e., if they are infinitesimal or large". This would be a strong advantage with respect to the restriction to the Gribov region, which has a strong dependence of those aspects. Can the authors make comments about the extension to other gauges and if this is compatible with BRST invariance?
It is true that our approach does not rely on the type of copies, as mentioned by the referee. However, the path integral formulation of our procedure requires that the Gribov copies one averages over correspond to extrema of the functional f[A,U] in the weight (4). In other words, the chosen gauge-fixing must have a formulation as an extremization problem. Only a few gauges are known to have this property. The most well-known one is the Landau gauge, which extremizes the functional (2). The only other example known to us is the class of Curci-Ferrari-Delbourgo-Jarvis gauges, which have been studied in terms of a similar averaging procedure as the one proposed here in Serreau et al. Phys. Rev. D 89 (2014) 125019. Unfortunately, the case of linear gauges does not fulfil this requirement. We have added a discussion on this point.
As for the question of the BRST symmetry, the standard BRST symmetry is softly broken by the averaging procedure (non flat weight over the Gribov copies). We have not identified a modified BRST symmetry in the present case.
R: My general comment is that the authors focus on the averaging procedure to eliminate Gribov copies, but mostly do not make any reference to what has been done using the restriction to the Gribov region" method. It would be beneficial for their work to connect their results with the other perspective in some way.
We agree with the referee. We have added some comments in this direction.
Referee: It is not clear to me how eq.(4) is defined for configurations that live on the so-called Gribov horizon. Such configurations represent zero modes of the operator F and, therefore, the denominator seems to be ill-defined. The authors could provide a comment on that.
We have added a discussion about this point (footnote 1).
R: The parameter ζ introduces a mass term for the Faddeev-Popov ghosts and the authors claim that it should not be taken as a problem. However, in the standard formulation of the Landau gauge, mass terms for the ghosts are forbidden by the Ward identities. In the standard (Refined) Gribov-Zwanziger construction this remains true. Can the authors comment about that and explain how to reconcile such things? Or am I missing something?
We are not completely sure what the referee means here. The point is that the Ward identities he/she refers to are those of the standard formulation of the Landau gauge, i.e., if we understand correctly, those of the Faddeev-Popov gauge-fixing. But symmetry identities are specific to each particular gauge-fixing and may not apply to others. For instance, a gluon mass term would also be forbidden by the standard Slavnov-Taylor identities. The construction we propose here is a different formulation of the Landau gauge, which comes with its own symmetry identities. In particular, the shift symmetry cbar -> cbar + constant, which forbids a mass term for the ghosts in either the FP or the GW constructions, is simply not a symmetry of the gauge-fixing proposed here.
R: The gluon-mass parameter is associated with the parameter β which is akin to a gauge parameter. This is very confusing to me. Are the authors claiming that the mass parameter that is generated by averaging over Gribov copies is gauge dependent and therefore can be taken to any value? Does it mean that β cannot enter correlation functions of gauge-invariant correlators?
As explained below Eq. (3), gauge-invariant observables are clearly independent of beta (and zeta) by construction. In a systematic expansion scheme (like perturbation theory), we expect this property to be respected order by order.
As for the gluon mass discussed here, it is not a gauge invariant quantity. We see no reason why it should be gauge independent. This, however, does not imply that it can take any value. For instance, as the phase diagram in Fig 2 shows, the value of the generated mass (only in the “massive” phase) is bounded.
Thinking about this point, we have realized that there may be an ambiguity in what one understands by “mass generation”. We have added some comments to clarify these points.
R: Can the authors connect the mass parameter that they obtain to the so-called Gribov parameter which is generated in the standard elimination of infinitesimal Gribov copies in the Landau gauge by the restriction of the path integral to the Gribov region?
No, we do not know how to do such a connection in a precise sense. We only have a qualitative intuition that our procedure favors configurations similar to those which are selected by the GZ restriction, i.e. near the first Gribov horizon. Indeed, the first horizon is enhanced by the denominator in (4) whereas higher Gribov regions are supressed by the exponential weight. It is certainly very hard to make a precise connection between the present proposal and the GZ procedure beyond this qualitative level, simply because these are in fact different gauge-fixing procedures. Relating the dimensionful parameters of these two to one another is certainly nontrivial. We have added a discussion on this point.
R: Below eq.(4), the authors say that F represents the FP operator in the Landau gauge and write F[A,U]=F[A^U] Can they explain what do they mean by this equality?
We simply mean that the function F[A,U] is, in fact, a function of A^U only and not of A and U separately. The revised version makes this clear.
R: It seems that this averaging method does not (strongly) rely on the choice of the Landau gauge and also on the type" of copies, i.e., if they are infinitesimal or large". This would be a strong advantage with respect to the restriction to the Gribov region, which has a strong dependence of those aspects. Can the authors make comments about the extension to other gauges and if this is compatible with BRST invariance?
It is true that our approach does not rely on the type of copies, as mentioned by the referee. However, the path integral formulation of our procedure requires that the Gribov copies one averages over correspond to extrema of the functional f[A,U] in the weight (4). In other words, the chosen gauge-fixing must have a formulation as an extremization problem. Only a few gauges are known to have this property. The most well-known one is the Landau gauge, which extremizes the functional (2). The only other example known to us is the class of Curci-Ferrari-Delbourgo-Jarvis gauges, which have been studied in terms of a similar averaging procedure as the one proposed here in Serreau et al. Phys. Rev. D 89 (2014) 125019. Unfortunately, the case of linear gauges does not fulfil this requirement. We have added a discussion on this point.
As for the question of the BRST symmetry, the standard BRST symmetry is softly broken by the averaging procedure (non flat weight over the Gribov copies). We have not identified a modified BRST symmetry in the present case.
R: My general comment is that the authors focus on the averaging procedure to eliminate Gribov copies, but mostly do not make any reference to what has been done using the restriction to the Gribov region" method. It would be beneficial for their work to connect their results with the other perspective in some way.
We agree with the referee. We have added some comments in this direction.

---

## Editorial Decision

published